# Determination of Tear Lipid Film Thickness Based on a Reflected Placido Disk Tear Film Analyzer

**DOI:** 10.3390/diagnostics10060353

**Published:** 2020-05-28

**Authors:** Pin-I Fu, Po-Chiung Fang, Ren-Wen Ho, Tsai-Ling Chao, Wan-Hua Cho, Hung-Yin Lai, Yu-Ting Hsiao, Ming-Tse Kuo

**Affiliations:** 1Department of Ophthalmology, Kaohsiung Chang Gung Memorial Hospital and Chang Gung University College of Medicine, Kaohsiung 83301, Taiwan; p520142001@cgmh.org.tw (P.-I.F.); fangpc@cgmh.org.tw (P.-C.F.); wen6530@cgmh.org.tw (R.-W.H.); a725157@cgmh.org.tw (W.-H.C.); d24758@cgmh.org.tw (H.-Y.L.); yuting1008@cgmh.org.tw (Y.-T.H.); 2Department of Laboratory Medicine, Kaohsiung Chang Gung Memorial Hospital and Chang Gung University College of Medicine, Kaohsiung 83301, Taiwan; tsaeling@cgmh.org.tw

**Keywords:** lipid layer thickness, tear film homeostasis, dry eye, meibomian gland dysfunction

## Abstract

This study aims at determining the thickness of the tear lipid layer (LL) observed from a placido-disc-based tear film analyzer. We prospectively collected reflections of placido-disk LL images using a tear film analyzer (Keratograph^®^ 5M, Oculus) from subjects with dry eye symptoms. The LL thickness (LLT) over the inferior half of the cornea was estimated with the use of interference color analysis and the preprocessing of images with and without ring segmentation were obtained and analyzed. Moreover, LLTs before and after 1 h of applying topical ointment (Duratears, Alcon) were compared to validate the estimation of LLT. Our results suggested that the tear LLT can be assessed using a placido-disk-based tear film analyzer and interference color analysis. We verified a high correlation between non-segmented and segmented LL images and estimated LLT increase after applying ointment. In addition, we concluded that LLT can be evaluated by direct interference analysis without segmentation preprocessing.

## 1. Introduction

Tears, the lubricant of the ocular surface, is composed of three distinct layers including lipid, aqueous, and mucins. [1] Dysfunction of any layer of the tear film will lead to dry eye disease (DED), which affects approximately 50% of the population worldwide. [2] In 2017, the Tear Film and Ocular Surface Society in the Dry Eye WorkShop (TFOS DEWS) II redefined DED as a multifactorial disease characterized by a loss of homeostasis of the tear film. Evaporative dry eye as a result of meibomian gland dysfunction (MGD) is more common than aqueous-deficient dry eye. [3] MGD is a chronic, diffuse malfunction of the meibomian glands characterized by terminal duct obstruction. [4] The meibomian gland secretes meibum which is the essential ingredient of the tear lipid layer (LL) for the prevention of tear film evaporation. [5]

LL performance is an important marker for MGD assessment. Guillon used Tearscope Plus (Keeler, Windsor, UK) to grade interference patterns of LL for determining tear film stability before contact lens fitting. [6] In addition, Goto and Tseng compared LL between normal subjects and patients with lipid tear deficiency by a DR-1 grading system (Kowa Co., Nagoya, Japan). [7] The DR-1 camera clearly showed the interference image of tear LL without background noise from iris color. Moreover, Eom et al. used LipiView Ocular Surface Interferometer (TearScience Inc., Morrisville, NC, USA) to compare LLT between normal subjects and MGD patients. [8] LipiView is a sophisticated instrument that can simultaneously analyze LLT, blinking rate, and capture meibography images. However, the cost-effectiveness of LipiView in medical investment is of concern and cannot be adopted at most clinics, therefore it is difficult to be widely applied in the dry eye population.

In recent years, a placido disk-based tear film analyzer (Keratograph^®^ 5M, Oculus; K5M), incorporating reflection topography and several kinds of tests for assessing dry eye, including tear dynamic LL examination, non-invasive keratograph break-up time (NIKBUT), tear meniscus height (TMH), has been used in the evaluation of DED patients. Although K5M can assess dynamic LL with blinks, it analyzes the LL qualitatively instead of quantitively. In addition, the effect of the intrinsic dark ring background of K5M on evaluating LL performance has not been clearly elucidated.

Therefore, the aim of this study was to establish and verify a novel, alternative standardized method to quantify the LLT based on the K5M system. Our main focus was to bridge the gap between the qualitative assessment of LL and automated LLT quantification of this widely-used tear film analyzer.

## 2. Materials and Methods

### 2.1. Subjects

A total of 28 subjects were prospectively enrolled in this study, which was part of an investigation of ocular microbiota for DED. All the subjects were enrolled at the corneal department of Kaohsiung Chang Gung Memorial Hospital (CGMH) between 1 February 2018 and 31 August 2018. Informed consent was obtained from each subject, and all procedures adhered to the Declaration of Helsinki and the statement of the Association for Research in Vision and Ophthalmology. This study was approved by the Committee of Medical Ethics and Human Experiments of CGMH, Taiwan (201600708B0, 28 July 2016). Subjects with dry eye symptoms were included, and those younger than 20 years old or older than 70 years old, those who had punctate keratitis, clinically apparent ocular inflammation, glaucoma under medical treatment, had undergone ocular or eyelid surgery less than 6 months, had diabetes mellitus, or were pregnant were excluded. The experimental framework of this study was summarized in a flowchart (Figure 1).

### 2.2. Clinical Tests

At the first visit, each subject received baseline clinical examination including a DED questionnaire, TMH, NIKBUT, and fluorescence staining slit-lamp examination. [9] All participants filled out the Ocular Surface Disease Index questionnaire (OSDI) for evaluation of their subjective ocular symptoms. All subjects were asked to stop applying any kind of eyedrop for 1 day and eye ointment for 2 days prior to the second visit, where LL examination was arranged. On the second visit, after the primary LL assessment, topical ointment (Duratears ointment, Alcon, Belgium) was applied for each subject immediately. One hour later, this LL examination was repeated to detect the post-lubricant LL performance. All examinations were performed by masked examiners for each subject.

### 2.3. Examination of Dynamic Tear Lipid Layer

A tear film analyzer, K5M, was used to detect the dynamic lipid film performance in an examination room with a constant temperature and humidity air conditioning system. The temperature and humidity of the examination room were controlled at 24 °C in the summer and 20 °C in the winter with 55% relative humidity. Each subject was instructed to have his or her head placed steadily on the head-chin rest of the instrument, looking attentively at the center of the placido ring illumination, and blinking naturally during the examination (Figure 2A). The white light projected onto the ocular surface and the interference colors of the moving lipid tear film were observed (Figure 2B). The examiner must focus on the lipid film of a subject first and then record the video under 1.4× magnification to obtain a qualified interference image. Generally, four to five continuous blink cycles were recorded. The initial blink cycles were excluded since the image is often defocused and requires fine-tuning at this point.

### 2.4. Selection of Tear Lipid Layer Image

For each subject, one blink cycle (Appendix A) of dynamic lipid tear film examination was selected according to the following criteria for subsequent image analysis. The selected blink cycle must meet the film quality criteria: (1) The focus must be sharp enough to visualize interference colors and movement of LL during the whole opening blink cycle (2) The placido ring reflection of the inferior half of the cornea must always be visible (3) The duration of the blink cycle must be over 2 s. Once the first qualified blink cycle from each subject was selected, the image sequences of this cycle were extracted from the decomposed video with a time interval of 0.05 s (Figure 2C and Appendix A) by an internet multimedia freeware, PotPlayer v1.7.13963 (Daum Communications Corp., Kakao, Korea). The image sequence upon eye-opening was defined as the initial image frame (*t* = 0 s) for standardization. A total of 41 decomposed image sequences were obtained from the beginning image frame (*t* = 0 s) to the final image frame (*t* = 2 s).

### 2.5. Region of Interest and Ring Background Segmentation

The standard procedures for the region of interest (ROI) selection and ring background segmentation were shown in Appendix A. ROI of each image frame was obtained by using the selection tool of ImageJ (1.42q, Wayne Rasband; National Institute of Health, Bethesda, MD, USA). In brief, after removing the region within the central 2 placido rings, the reflection of the placido ring of the inferior half of the cornea was selected as the pre-segmented ROI (Figure 3A,C; Part A in Appendix A) because lipid film generally spreads over the inferior cornea while the superior cornea is sometimes partially covered by the upper eyelid and eyelashes. The maximal entropy-based thresholding algorithm [10] was adopted to remove the dark ring background in the pre-segmented ROI and extract the bright ring segment to obtain the post-segmented ROI (Figure 3B,D; Part B in Appendix A).

### 2.6. Estimation of Lipid Layer Thickness

According to Hwang et al. [11] LLT can be estimated by referring each pixel’s color (red, green, and blue scales; RGB) to a look-up table (Appendix A) and approximated by the principle of the nearest Euclidean distance:(1)$$Euclidean\;distance=\sqrt[{2}]{{\left(r_{i}-R_{j}\right)}^{2}+{\left(g_{i}-G_{j}\right)}^{2}+{\left(b_{i}-B_{j}\right)}^{2}},$$
in which (*r_i_*, *g_i_*, *b_i_*) is a point of an ROI image, and (*R_j_*, *G_j_*, *B_j_*) is a reference point in the look-up table.

The look-up table of LLT is a 3-dimensional curve in a 3-D Cartesian coordinate system (*R* axis, *G* axis, and *B* axis), and LLT is determined with 3 variables (*R* scale, *G* scale, and *B* scale). There are known RGB scales for the reference points that correspond to the LLTs in the look-up table. After identifying the reference point in the look-up table with the minimal distance to a point (pixel) in an ROI image, its corresponding LLT was assigned as the LLT of the point in the ROI image. The procedure to obtain the LLT of ROI was summarized in Appendix A. In brief, the look-up table was pre-loaded into the Excel worksheets. After splitting an ROI image into 8-bit RGB images, the LLT at each pixel coordinate can be obtained after loading the split images into the above Excel worksheets and running the Macro program named as LLT. The average LLT was calculated by summation of LLT at each pixel coordinate in the ROI divided by the pixel number of ROI and implemented for ROIs of both pre- and post-segmented images. To obtain pre- or post-segmented LLTs in ROI without burden, an application software (Appendix A) was developed for saving time and reducing mistakes from these procedures shown in Appendix A. The instruction for operating the software is shown in Appendix A.

### 2.7. Statistical Analysis

An Excel plug-in program, Analysis ToolPak, was used to perform the paired t-test for examining the difference of average LLTs between pre- and post-segmentation images, and between pre- and post-lubrication images at specific time points. In addition, Pearson coefficient r was used to examine the correlation of average LLTs between pre- and post-segmentation images. Moreover, a general linear model was used to predict the post-segmentation LLT by pre-segmentation LLT at time points with a high correlation. Intraclass correlation coefficient (ICC), which was assessed by a free ICC-reliability calculator (Mangold Lab Suite version 2015, program version 1.5, Arnstorf, Germany), was used to examine the reliability of the LLT quantification method. Statistical significance was recognized at *p* < 0.05.

## 3. Results

### 3.1. Subjects

A total of twenty female subjects and eight male subjects participated in this study (Table 1). The mean age of these participants was 40.6 ± 15.0 years old. The right eye of each subject was used for assessing tear film homeostasis. The OSDI score, NIKBUT, and central TMH of these subjects were 33.1 ± 26.4, 8.3 ± 5.2 s, and 0.23 ± 0.07 mm, respectively.

### 3.2. Dynamic Lipid Layer Shown on the Placido Ring Tear Film Analyzer

Lipid tears are spread over the ocular surface from the most inferior cornea to the central cornea following the opening cycle of blinks (Appendix A). Figure 2C demonstrated the image sequences with a time interval 0.05 of s decomposed from a dynamic LL assessment using the K5M tear film analyzer. The colorful LL was clearly visible in the bright white ring regions as opposed to the dark brown ring regions of the placido ring reflection background. For the same subject shown in Figure 2C, the LL spread rapidly upward within 0.5 s, the tip of upward LL clearly spread through the innermost white circle at 0.7 s, and LL remained static and across the entire cornea surface after 1.10 s. The pre- and post-segmentation LL images of subjects no. 1 and 2 were shown in Figure 3. After segmentation, the brown placido ring background was totally removed, including some areas with eyelash coverage. In addition, a small area of bright white ring over the boundary was also trimmed under maximal entropy segmentation to obtain a purified ROI of post-segmentation.

### 3.3. Average Lipid Layer Thickness before and after Segmentation

The mean LLTs at different time points for cases no. 1 and 2 were estimated from the respective ROI of pre- and post-segmented images (Figure 4A). We found the dynamic LLT of pre-segmentation was thicker than that of the post-segmentation for the 2 subjects, whereas the LLT had the same trend between pre- and post-segmentation for the same case. In the following, a sequence of pre- and post-segmented images from the first 12 subjects (case no. 1 to no. 12) were obtained at various selected time points. At time points 0.05 s, 0.10 s, 0.25 s, 0.5 s, 1.0 s and 2.0 s (Figure 4B), the mean LLT before placido ring segmentation were 55.8 ± 27.5 nm, 58.2 ± 27.8 nm, 58.8 ± 25.4 nm, 60.2 ± 25.4 nm, 59.7 ± 25.0 nm and 58.9 ± 25.1 nm, respectively, while those after segmentation were 46.3 ± 21.6 nm, 53.8 ± 23.0 nm, 49.9 ± 19.3 nm, 51.8 ± 19.9 nm, 53.6 ± 22.6 nm and 44.8 ± 13.9 nm, respectively. There were no significant differences between pre- and post-segmentation LLTs at different time points except at 2.0 s, of which pre-segmentation LLT was significantly thicker (*p* < 0.01) than post-segmentation LLT.

However, the correlation between pre- and post-segmentation of LLTs was significantly positive at different time points except 0.05 s (*p* > 0.2) (Figure 5). The correlation increases with time, which means that lipid tears converge from turbulence to stagnation. Thus, the LLT over the white ring region is towards more constant when the eye is opened longer between the time points 0.1 s and 2.0 s.

Due to significantly high correlation between pre- and post-segmentation LLTs at 0.5 s, 1.0 s, and 2.0 s from the above analysis, the LLTs at these time points were used to obtain the precise correlation, which was further verified by including all subjects. Thus, the transformed LLT based on the pre-segmentation LLT for predicting post-segmentation LLT by general linear model was firmly established for these points (at 0.5 s, the transformed LLT = pre-segmentation LLT × 0.65 + 8.73; at 1.0 s, the transformed LLT = pre-segmentation LLT × 0.58 + 11.27; at 2.0 s, the transformed LLT = pre-segmentation LLT × 0.55 + 11.01) (Figure 6). For simplicity, we could quickly estimate the transformed LLT = pre-segmentation LLT × 0.6 + 10 for these time points.

### 3.4. Verification of LLT Detection on K5M Tear Film Analyzer

To verify the detection of LLT based on the K5M tear film analyzer, the intraclass correlation coefficient (ICC) was calculated to assess the repeatability of LLT detections in 2 repeated ROI selections and 2 consecutive opening blink cycles for each subject. We found ICCs to be very high at the 3 representative time points (Figure 7), which means that this method has excellent reliability.

Moreover, we compared the LLTs before and after applying an oil-based lubricant for all subjects, as the LLT should increase after the intervention if our proposed method works. At three different time points, the transformed LLTs before lubrication were 48.1 ± 23.8 nm, 49.6 ± 23.7 nm, and 50.1 ± 24.0 nm, while those after lubrication were 68.4 ± 29.1 nm, 71.4 ± 30.9 nm, and 73.7 ± 31.6 nm, respectively (Figure 8A). The LLT after lubrication were significantly increased. Moreover, after lubrication, all subjects had an increase in LLT at 0.5 s, and only 1 participant (case no. 5) showed a decrease in LLT at timepoints 1.0 s and 2.0 s (Figure 8B).

## 4. Discussion

The essential role of LL in the tear film has attracted attention following better understanding of the pathophysiology of DED. In addition to DED, LL is also affected by many other ocular diseases [12,13,14]. Therefore, many commercially available instruments were developed in recent years for better evaluation of LL. Among these instruments, the K5M tear film analyzer has become increasingly -popular due to its comprehensive functions for DED assessment. However, the K5M assessment for LL only provides qualitative instead of quantitative analysis and has an inherent drawback due to the dark placido ring background. In this study, we tried to overcome these limitations and estimate the LLTs of patients from the extracted images using K5M tear film analyzer. We found the estimated LLTs of pre- and post-placido ring segmentation was highly correlated at 0.5 s in the opening phase of the blink cycle. Based on our general linear model, images extracted from the tear film analyzer at time points 0.5 s, 1.0 s, and 2.0 s may be directly used to estimate LLT in clinical practice without the need for image preprocessing with placido ring segmentation. By analyzing two repeated ROI selections and two consecutive blink cycles, we found the estimated LLT to have excellent repeatability. In addition, the estimated LLT was further verified by comparing the pre- and post-lubrication LLT of the same subject; the obtained LLT was significantly increased after oil-based lubrication.

Some commercially available LL analyzers have been used to assess the LLT. Guillon mounted Tearscope Plus on a slit-lamp biomicroscope to observe the lipid interference image formed on the central corneal region. They proposed five grading patterns: open meshwork, closed meshwork, wave or flow, amorphous and color fringe [6], and suggested that thicker LL had amorphous and closed meshwork patterns, while on the contrary, thinner LL had color fringes and open meshwork patterns. However, this tearscope-based system only offered a four-step resolution in determining LLT. The DR-1 camera clearly focused on the interference image of LL of tear film without dark background. Yokoi et al. used DR-1 camera to obtain the interference color image of LL and proposed a five-step grading system (grade one to five). [15] Goto and Tseng [7] used DR-1 to analyze kinetic changes of tear LL for exploring LL spread pattern and speed and estimating LLT based on a look-up table published by Korb [16]. They established that healthy subjects had a horizontal LL spread pattern and slower LL spread compared to the lipid tear deficiency group. [7] In addition, they also found that the Yokoi grading system cannot effectively differentiate lipid tear deficiency patients from healthy subjects. Moreover, they further adopted an ROI of inferior 2/3 cornea and the time point of 0.4 s to estimate LLT, and identified the mean estimated LLTs of 44 nm and 75 nm in lipid tear deficiency patients and healthy subjects, respectively. In recent, spectral domain-optical coherence tomography (SD-OCT) has had an impact in the research setting of tear film dynamics. [17] However, even with an ultra-high-resolution SD-OCT system, the axial resolution is only 1 μm (1000 nm). Therefore, a further technical improvement in a SD-OCT system may be needed for the measurement of LLT, which was estimated to be about 15 to 160 nm. [18]

LipiView system adopts an ROI of about 1/3 of the inferior cornea and provides real-time visualization of the LL to evaluate the dynamic response of lipids to blinking. The average, maximal, and minimal LLT within the assessment period were simultaneously obtained through built-in automatic analysis software. It is a sophisticated LL analyzer but the cost-effectiveness consideration precludes many medical centers in Taiwan from making this medical investment. Eom et al. used LipiView to compare LLTs between patients with obstructive MGD and normal subjects [8] and found LLT was negatively correlated with upper and lower meibomian gland losses in both groups. In addition, the average LLTs of MGD and normal subjects were 54 nm and 65 nm, respectively. In this study, we used K5M to assess the subjects before and after instillation of eye ointment and found the mean pre- and post-transformed LLTs at timepoints 0.5 s were 53 nm and 63 nm, respectively. Upon comparison with the study from Eom et al., besides the characteristic differences in subjects, the larger ROI of inferior 1/2 of the cornea based on K5M included more dynamic changes of LL but obtained thinner LLT than that of the LipiView system.

Goto and Tseng used the DR-1 system and selected timepoint 0.4 s to determine the LLT of each subject because they found that the LL usually stabilizes after 0.4 s. We used K5M to observe the lipid film and found that LL spread tends to stabilize after 0.5 s (Figure 2C) and becomes nearly static after 1 s. Lipid film deficiency in the superior ROI and uneven distribution was observed in both DR-1 [19] and K5M. The concern of reducing discriminant sensitivity due to LLT decreasing with time after blinking was raised for some subjects under our ROI setting. Therefore, we recommend the image frame at timepoint 0.5 s as the standard of the K5M tear film analyzer for determining representative LLT per blink cycle. Fukuoka used a topical lubricant Diquas (Santen, Japan) and evaluated LLT change by LipiView and found the lubricant effect on healthy subjects could last up to one hour, of which LLTs were increased by approximately 15 nm one hour after instillation [20]. We found the mean pre-segmentation LLT and the transformed LLT according to the 0.5-s image frame of K5M increased by about 20 nm and 10 nm respectively at one hour after ointment application.

Comparing the pre- and post-segmentation LLTs at different time points (Figure 4B), the median LLTs of pre-segmentation only changed by a little over time, whereas those of post-segmentation were vibrated before 0.5 s after eye-opening and then decreased with time after 0.5 s. After interference color look-up table conversion, the dark ring background may cause LLT overestimation. The falsely high estimation of LLT may cause significantly higher LLTs before segmentation than those after segmentation at timepoint 2.0 s (*p* = 0.0047). Moreover, the positive correlation of the LLT between pre- and post-segmentation increased gradually with time (Figure 5), which further confirmed that LL image before timepoint 0.5 s was not stable and unsuitable for LLT conversion between pre- and post-segmentation images. Furthermore, we calculated the transformed LLT based on the pre-segmentation LLT for predicting post-segmentation LLT by the general linear model (Figure 6). The obtained formula could help us obtain the post-segmentation LLT without requiring spending extra time in image segmentation. In addition, the newly developed software could shorten the time considerably when obtaining the LLT from a lipid film image after the selection of ROI estimation.

Our novel quantification method of LLT based on K5M may provide a practical, time-efficient, cost-effective protocol for DED assessment in routine clinical practice. In the reliability analysis, ICCs in the estimated LLTs obtained from two repeated ROI selections of the same image and from the same time point of two blink cycles indicated excellent repeatability of the method (Figure 7). The LLT detected at 0.5 s had the highest inter-blinks ICC (Figure 7B), therefore, the image extracted at this time point may be the ideal choice when selecting the representative LLT of a subject. In addition, we also compared the LLTs before and after applying an oil-based lubricant for all subjects. After lubrication, only 1 participant (case no. 5) had LLTs decrease at timepoints 1.0 s and 2.0 s (Figure 8B). Her LLT was slightly decreased from 48 nm to 46 nm at 1.0 s and from 48 nm to 47 nm at 2.0 s, respectively. Although her dry eye symptoms were moderate in severity (OSDI score 27.1), she was young (29 years old), had normal tear rupture (NIKBUT 27.1 s), and tear secretion (TMH 0.20 mm) and had no systemic disease. We speculate that the higher tear turnover rate may be the reason why the lubricating effect faded faster than other objects. Interestingly, her LLT was increased at 0.5 s as well, which further suggested that this timepoint was influenced by tear turnover the least and may be the best timepoint to obtain a reliable LLT.

There were some limitations to this study. First, subjects with dry eye symptoms but without aqueous deficiency were still included in this study for the purpose of enrolling lipid deficiency subjects. Among these subjects, OSDI scores of 6 subjects were less than 13, NIKBUT of 7 subjects were longer than 10 s, and 8 subjects did not match the criteria of DED proposed by TFOS DEWS II. However, the aim of this study was to elucidate the influence of placido ring interference on LLT estimation and verify the innovative formula of estimating LLT based on K5M. Thus, pre-classification into different DED types for these subjects were not considered. Second, there were no other LL analyzers in this study to assess the inter-system reliability. Further studies may be considered in applying our quantification method in different LL analyzers. Despite different LL analyzers using the same theory of color interference, these instruments may differ in ROI settings and image frames for LLT determination. Therefore, the interpretation of LLT should be established individually for each instrument, including the ROI and algorithm for LLT determination. Third, similar to other LL analyzers, the LLT of K5M was estimated by detecting each pixel’s color intensity in the interference image, which could be influenced by the refractive indices of tear lipid and aqueous layers and specular angle. In addition, we cannot really know the resolution of the method. Fourth, as microfluctuations may occur during the dynamic ocular examination, making it nearly impossible to guarantee the reference points may remain the same on the corresponding tear film area or points ((*x*, *y*) coordinates) in a complete blink cycle. However, the LLT of a specific timepoint was estimated by averaging data about 500,000 points or pixels from an ROI image, therefore the deviations of estimated LLTs caused by the ROI selection or natural ocular microfluctuations were very small. High repeatability coefficients of estimated LLTs in inter-ROI selections and inter-blinks had further confirmed our surmise. Fifth, technicians were unable to complete the quantification of LLTs in one step when using our established technique. The user must take time to operate PotPlayer to extract the representative image from a video file and select the ROI with ImageJ. The most time-consuming step is to determine the LLT by splitting an ROI image into RGB images in order to be able to refer to a look-up table. However, we have simplified this step through developing an application software. Although the K5M approach may require a little more time than a specialized LL analyzer like LipiView, it can provide an alternative approach to quantify LLT, which is especially beneficial for a clinician or researcher without sufficient budget.

## 5. Conclusions

In conclusion, the LLT can be estimated by using a placido-disk based tear film analyzer with interference color analysis. Since a high correlation between segmentation and non-segmentation in image frames after timepoints 0.5 s was observed, we propose that direct color analysis without segmentation preprocessing can be used to evaluate the LLT through a simple linear formula. The non-segmented image at 0.5 s is the most reliable timepoint to obtain the representative LLT for a patient with dry eye symptoms. This proposed method will provide a more cost-effective approach for LLT assessment in daily practice.

## Figures and Tables

**Figure 1 diagnostics-10-00353-f001:**
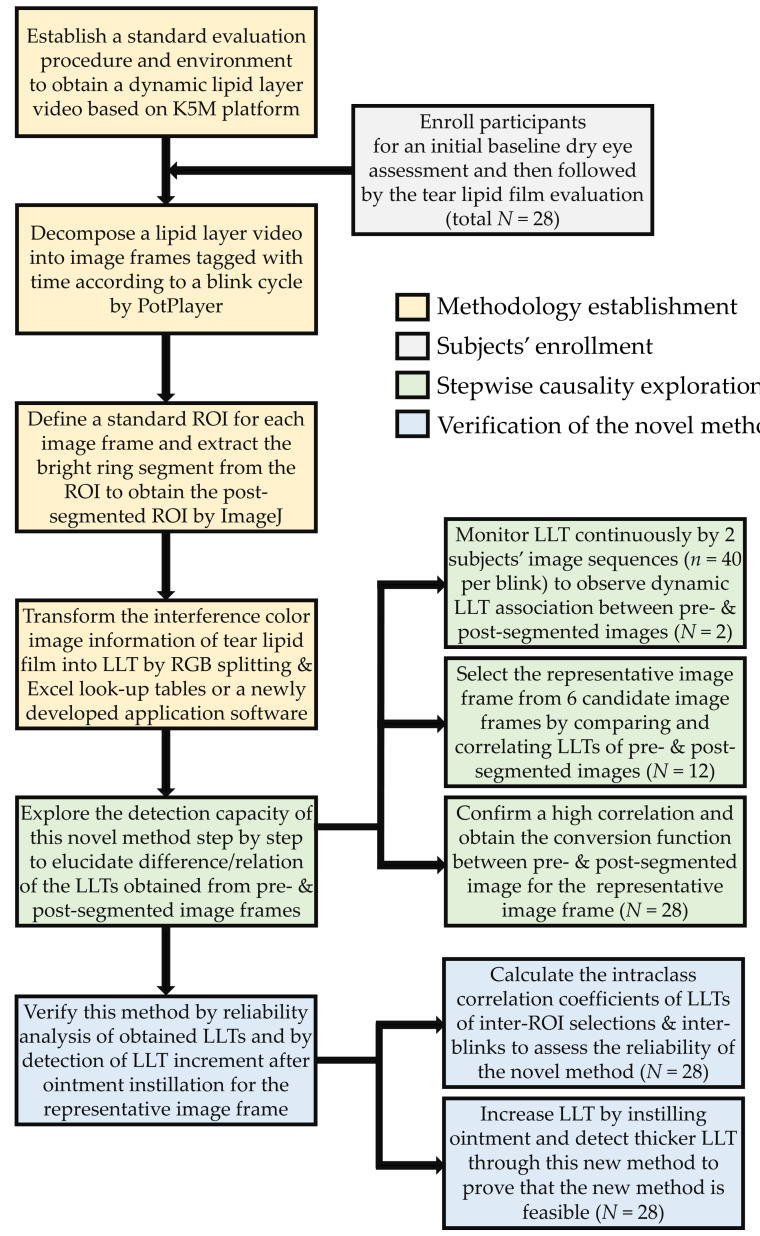
The flowchart of the experimental framework of this study. This experiment consisted of 4 parts, including the methodology establishment based on the K5M tear film analyzer, enrollment of subjects with dry eye symptoms, exploration of the causality of pre- and post-placido-disk ring segmented images for further selection of the representative image frames, and verification through examining the repeatability of acquired LLTs in two repeated ROI selections and two different blink cycles in a test and detecting the increment of LLT one hour after ointment instillation for each subject. K5M = Keratograph^®^ 5M, Oculus; ROI = region of interest; LLT = lipid layer thickness; RGB = red, green, and blue scales.

**Figure 2 diagnostics-10-00353-f002:**
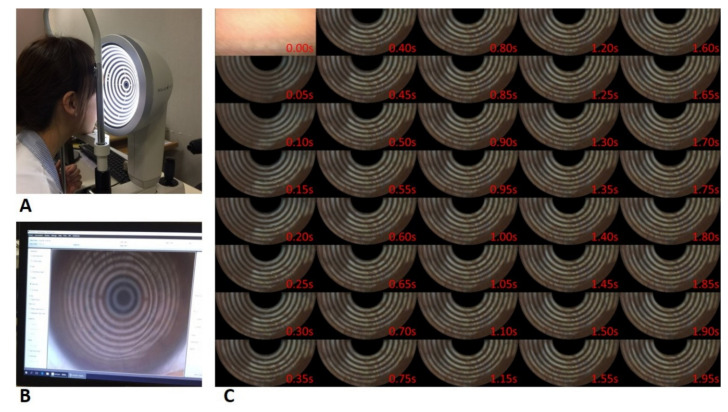
Lipid tear film evaluation by a tear film analyzer, Keratograph 5M. (**A**) Assessment of a subject with the instrument; (**B**) A video with interference colors of the moving lipid tear film is recorded under the white ring light illumination; (**C**) Interference image sequences of lipid layer decomposed from a video record of lipid tear film examination with a time interval of 0.05 s. These interference images reveal dynamic lipid tear film spreading on the ocular surface, especially on the white ring regions.

**Figure 3 diagnostics-10-00353-f003:**
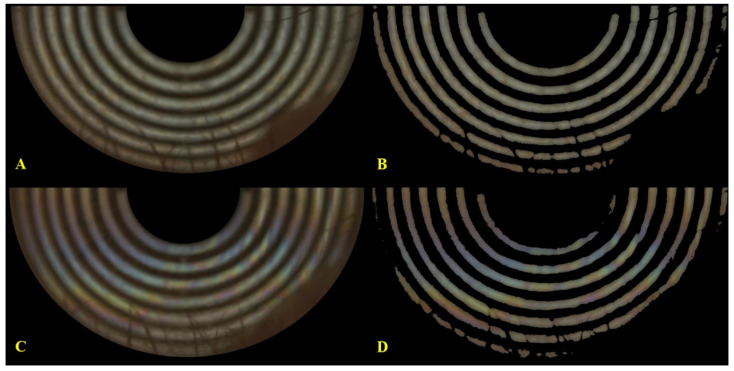
The pre- and post-segmentation interference lipid tear film images of the right eyes of two subjects. (**A**,**C**) the lipid tear image of patient no. 1 and 2, respectively, before the segmentation processing; (**B**,**D**) the lipid tear image of patient no. 1 and 2, respectively, after the image segmentation processing. All images showed shadows of eyelashes and inferior medial eyelid over the inferior area. Comparing pre- and post-segmentation images for the same subject, the bright ring showed a thinner width and were trimmed at some blurred area after segmentation. Furthermore, the post-segmented image showed clearer and sharper presentation than the pre-segmented image.

**Figure 4 diagnostics-10-00353-f004:**
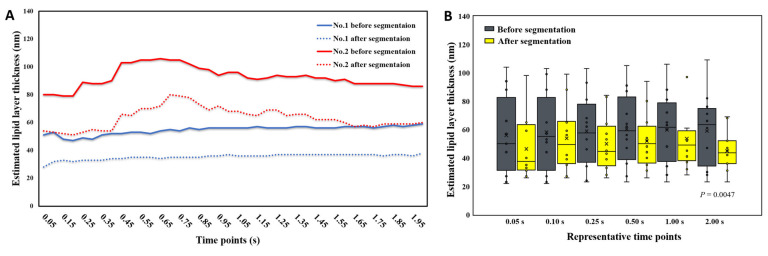
Comparison of the estimated LLT at different time points before and after image segmentation. (**A**) the serial estimated LLT with time interval 0.05 s from pre- and post-segmented images of patient no. 1 and 2, respectively; (**B**) the estimated LLT at different time points from pre- and post-segmented LLTs shown by boxplots (*N* = 12).

**Figure 5 diagnostics-10-00353-f005:**
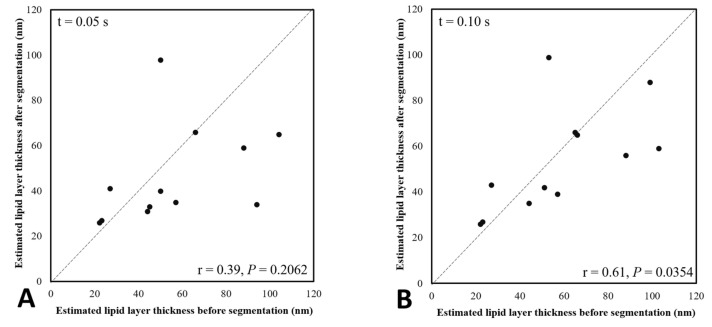
The correlation of the estimated lipid layer thickness between pre- and post-segmentation at different time points for the right eyes of the 12 subjects. (**A**) at 0.05 s; (**B**) at 0.1 s; (**C**) at 0.25 s; (**D**) at 0.5 s; (**E**) at 1.0 s; (**F**) at 2.0 s.

**Figure 6 diagnostics-10-00353-f006:**
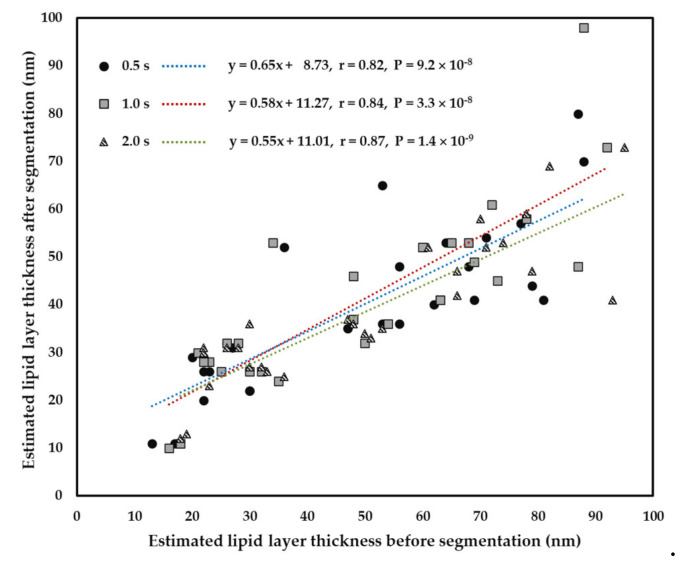
Transformed LLT based on the pre-segmentation LLT for predicting post-segmentation LLT by general linear model (at 0.5 s, the transformed LLT = pre-segmentation LLT × 0.65 + 8.73; at 1.0 s, the transformed LLT = pre-segmentation LLT × 0.58 + 11.27; at 2.0 s, the transformed LLT = pre-segmentation LLT × 0.55 + 11.01). Total number of subjects were 28 (*N* = 28).

**Figure 7 diagnostics-10-00353-f007:**
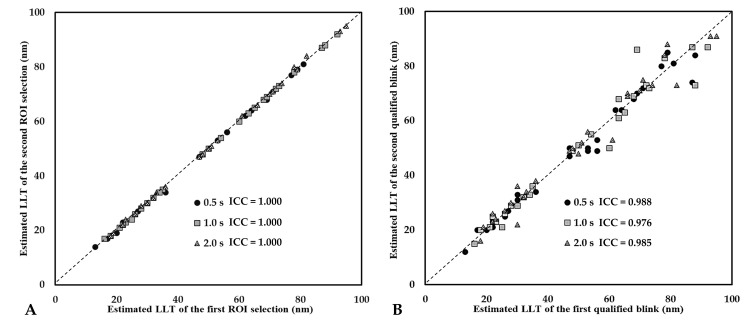
Non-segmentation images were used to assess the reliability of estimated lipid layer thickness for each subject at the representative time points, 0.5 s, 1.0 s, and 2.0 s (*N* = 28). (**A**) the reliability of estimated LLTs in two repeated ROI selections for the first qualified blink cycle; (**B**) the reliability of estimated LLTs of the first and second qualified blink cycle. ICC = intraclass correlation coefficient.

**Figure 8 diagnostics-10-00353-f008:**
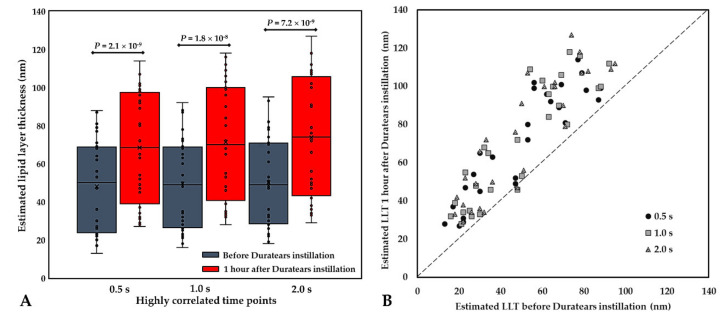
Using the non-segmentation images to assess the estimated lipid layer thickness before and 1 h after Duratears instillation at the highly correlated time points. (**A**) Comparison of the estimated LLT between pre- and 1-h post-ointment instillation at 0.5 s, 1.0 s, and 2.0 s in the opening phase of the blink cycle; (**B**) The change of the estimated LLT after ointment instillation for each patient at 0.5 s, 1.0 s, and 2.0 s in the opening phase of the blink cycle. Total number of subjects was 28 (*N* = 28).

**Table 1 diagnostics-10-00353-t001:** Demographic data of participants.

Baseline Characteristics	Data
No. of subjects, *N*	28
Sex, F:M	20:8
Eye, OD:OS	20:8
Age, median (range), years	41 (21–68)
OSDI *^a^*, median (range), scores	26.1 (6.3–95.8)
NIKBUT *^b^*, median (range), s	6.9 (1.0–24.0)
TMH *^c^*, median (range), mm	0.24 (0.9–0.35)

*^a^* OSDI = Ocular Surface Disease Index questionnaire; *^b^* NIKBUT = Non-invasive keratograph break-up time assessed by Keratograph^®^ 5M tear film analyzer, and the first non-invasive keratograph break-up time was used to represent this parameter; *^c^* TMH = Tear meniscus height examined by Keratograph^®^ 5M tear film analyzer, the tear meniscus height measured at the inferior middle eyelid was used to represent this parameter.

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
