# Peer review of "Determination of Tear Lipid Film Thickness Based on a Reflected Placido Disk Tear Film Analyzer"

_diagnostics, 2020, doi:10.3390/diagnostics10060353_

Round 1

Reviewer 1 Report

I read with interest the article by Pin-I Fu et al., attempting to determine the thickness of tear lipid layer observed from a placido-disc based tear film analyzer. However, several questions are raised in this manuscript that need to be addressed and clarified.

  1. The rationale of this study is rather vague. Please clarify the utily of this study.
  2. I cannot found a proper statistical analysis.
  3. No data of repeatability and reproducibility have been shown. It seems inconclusive and subjective.
  4. Some concerns are related to the diagnosis of dry eye: how you analyze the tear film? How were the environmental conditions?
  5. Bibliographic research seems incomplete. Please note that a number of non-invasive techniques for evaluating the lipid tear film are reported in the literature. For example,

"Fourier-Domain OCT Imaging of the Ocular Surface and Tear Film Dynamics: A Review of the State of the Art and an Integrative Model of the Tear Behavior during the Inter-Blink Period and Visual Fixation." Journal of Clinical Medicine 9.3 (2020): 668.

In sum, the article is interesting, but the interpretation may be incorrect. In my opinion, the paper at the moment is not ready for publication. Please provide the requested improvements.

Author Response

Q1 The rationale of this study is rather vague. Please clarify the utility of this study. 

A1 The K5M is a cost-effective, commercially available, and multifunctional tear film analyzer. However, in the assessment of lipid layer (LL), the analyzer could only provide a subjective LL qualitative examination result instead of an objective LL quantification. Our motivation was to establish and verify a novel, alternative standardized method to quantify the lipid layer thickness (LLT) based on the K5M system. We aimed to bridge the gap between qualitative assessment of LL and automated LLT quantification using this widely-used tear film analyzer. We have clarified the rationale of this study in the last paragraph of the introduction (Lines 54-56, page 2).

Q2 I cannot find a proper statistical analysis.

A2 We added a flowchart (Figure 1, page 3) to help readers understand the framework of this study. Moreover, to strengthen the statistical analysis, we added a reliability analysis (Lines 155-157, page 5) to verify the repeatability of LLT measurement for the representative timepoints of the first and second qualified opening blink cycle (Figure 7 and lines 223-232, page 9).

Q3 No data of repeatability and reproducibility have been shown. It seems inconclusive and subjective.

A3 We thank for the reviewer for this important comment. We have added the reliability analysis of the repeated measurement of LLT. We found that the repeatability is excellent at the representative time points, especially at 0.5 s (Figure 7 and lines 223-232, page 9. Lines 321-324, page 11).

Q4 Some concerns are related to the diagnosis of dry eye: how you analyze the tear film? How were the environmental conditions?

A4 In our clinical practice, we adhere to the consensus of TFOS DEWS II for diagnosis of dry eye. However, subjects with dry eye symptoms but without matching screening criteria (OSDI>13) or homeostatic markers (NIKBUT, ocular surface staining) were still included in this study for the purpose of enrolling lipid deficiency subjects. In addition, the aim of this study was to elucidate the influence of placido ring interference on LLT quantification and verify the innovative method of estimating LLT based on K5M. Thus, pre-classification into different DED types for these subjects were also not considered. We mentioned the above concerns in our study limitations (Lines 334-340, page 11).

A tear film analyzer, K5M, was used to detect the dynamic lipid film performance in an examination room with a constant temperature and humidity air conditioning system. The temperature and humidity of the examination room were controlled at 24°C in the summer and 20°C in the winter with 55% relative humidity (Lines 79-82, page 2).

Q5 Bibliographic research seems incomplete. Please note that a number of non-invasive techniques for evaluating the lipid tear film are reported in the literature. For example, "Fourier-Domain OCT Imaging of the Ocular Surface and Tear Film Dynamics: A Review of the State of the Art and an Integrative Model of the Tear Behavior during the Inter-Blink Period and Visual Fixation." Journal of Clinical Medicine 9.3 (2020): 668.

A5 We agree that spectral domain-optical coherence tomography (SD-OCT) has an impact in the research setting of tear film dynamics recently. [17] However, even with an ultra-high-resolution SD-OCT system, the axial resolution is only 1 mm (1000 nm). Therefore, the K5M analyzer may provide a higher resolution to further assess LLT (10~160 nm). We have mentioned the impact and limitation of SD-OCT in the measurement of LLT in the discussion (Lines 277-280, page 10).

Reviewer 2 Report

The paper by Fu, P-I is interesting and highly relevant for the field of ocular surface research. LipiView is an interesting instrument that can simultaneously analyze LLT, blinking rate, and capture meibography images. I have a few comments to improve the manuscript:

(1) This is a method and analysis paper. Why do not the authors explain the method more in detail?

(2) I also do not understand what is the necessity of having the Table 1 in the manuscript?

(3) Why Figure 3-5 only contain 12 patients? 

(4) To increase the readability from the researchers of the field, they should include a flowchart of their methodology.

Author Response

Q1 This is a method and analysis paper. Why do not the authors explain the method more in detail?

A1 We thank the reviewer for this comment. We have indeed shown the details of this novel method without reservation. If readers can take time to download all the supplementary files and implement LLT estimation according to the information step-by-step as described in the method, they will find it easy to measure LLT from the original lipid tear film image.

Q2 I also do not understand what is the necessity of having the Table 1 in the manuscript?

A2 We agree with this comment that the details of Table 1 are somewhat redundant. We have summarized the baseline characters of subjects in the revised Table 1 (Lines 165-170, page 7).

Q3 Why Figure 3-5 only contain 12 patients?

A3 This study consisted of an extensive but reliable exploration process. Initially, we analyzed the LLT of two subjects with decomposed images at high density time points (2 patients x 2 kinds of image x 40 time points). Then we increased the number of subjects and the included images were limited to six candidate time points (12 patients x 2 kinds of image x 6 time points) for selection of the representative time points to extract images from videos. Finally, all subjects were included to explore the correlation of segmented and non-segmented images at the three representative time points (28 patients x 2 kinds of images x 3 time points). We found that we could visually analyze the trend of the LLT change with our current sample size. Therefore, we adopted this principle to perform this study.

Q4 To increase the readability from the researchers of the field, they should include a flowchart of their methodology.

A4 We thank you very much for this valuable comment. We have presented the framework of this study in a flowchart (Figure 1, page 3) and believe the readability of this article will be greatly increased.

Round 2

Reviewer 1 Report

The revised version of the article written by Pin-I Fu et al. is now better than before.

However, there are still some important flaws. I'd like to help authors make the article worth publishing.

  1. The article reports “K5M analyzer may provide a higher resolution to further assess LLT (10~160 nm)”. The authors must indicate the proper reference for this feature and elucidate axial and transverse resolution of K5M. Please note “LLT can be estimated by referring each pixel’s color”. Please specify the limits of this approach.
  2. Please remove R2! Why did you use this coefficient?
  3. Repeatability analysis is necessary for all subjective measurements. For example, the selection of ROI. In fact, file 3 reports “Keep the inferior half circular region by Rectangle selection tool and Crop function in the Image menu.” This is very subjective! Poor standardized. Please provide repeatability coefficients of your data.
  4. How did you guarantee the analysis of the same tear film area or points [(x, y) coordinates] for comparison over time? Please specify the limits of this approach.
  5. Overall, this technique is full of steps and very complex. Please specify the limits of this approach.
  6. Please specify the ‘macro’ formula in Excel (in another supplementary file) and how you got there.

Author Response

Q1 The article reports “K5M analyzer may provide a higher resolution to further assess LLT (10~160 nm)”. The authors must indicate the proper reference for this feature and elucidate axial and transverse resolution of K5M. Please note “LLT can be estimated by referring each pixel’s color”. Please specify the limits of this approach.

A1

Thank you for pointing this out. We have cited a reference about the range of LLT in human tears and modified the statement (Lines 289-291, page 11). Currently, there is not enough information available to verify the axial and transverse resolutions of K5M, because K5M is a placido disk topography-based tear film analyzer, designed originally for qualification of lipid layer analysis instead of quantification. We agree that the LLT of K5M was estimated by detecting each pixel’s color intensity in the interference image, which could be influenced by the refractive indices of tear lipid and aqueous layers, and specular angle. We have added the limitations to the discussion in the revised manuscript (Lines 357-360, page 12).

Q2 Please remove R2! Why did you use this coefficient?

A2

We have modified all statements with R2, and taken out R2 in the revised manuscript.

Q3 Repeatability analysis is necessary for all subjective measurements. For example, the selection of ROI. In fact, file 3 reports “Keep the inferior half circular region by Rectangle selection tool and Crop function in the Image menu.” This is very subjective! Poor standardized. Please provide repeatability coefficients of your data.

A3

We agree that subjectivity exists in the ROI selection step. However, the LLT at a specific timepoint was estimated by averaging the data from about 1000 x 500 pixels (500,000 points) of a ROI image. Therefore, we may find extremely high repeatability coefficients in two repeated ROI selections for the same image. We have amended the repeatability analysis as two parts to obtain the repeatability coefficients of repeated ROI selections and two consecutive blinks (Figure 7, page 9; lines 231-234, page 9; lines 267-269, page 10; lines 332-335, page 12).

Q4 How did you guarantee the analysis of the same tear film area or points [(x, y) coordinates] for comparison over time? Please specify the limits of this approach.

A4

Thank you for this suggestion. We cannot guarantee the reference points selected on tear film area or points may remain the same over time because naturally, microfluctuations may occur during the dynamic examination. Failure to establish a highly correlated analysis may be possible even when we compare the same point (pixel) under a system with automatic selection and registration of ROI, especially when dealing with images of high resolution or pixel numbers. Our approach was to estimate the average LLT of a subject from a ROI image, which had near half million pixels. Thus, we found the repeatability of the inter-blinks to be very high. We have added this limit in the discussion of the revised manuscript (Lines 360-366, page 12).

Q5 Overall, this technique is full of steps and very complex. Please specify the limits of this approach.

A5

We agree that this technique cannot be completed in one step by a technician. The user must take time to operate PotPlayer to extract the representative image from a video file and select the ROI with ImageJ. The most time-consuming step is to determine the LLT through splitting an ROI image into RGB images in order to be able to refer to a look-up table. However, we have simplified this step through developing an application software. Although the K5M approach may require a little more time than a specialized LL analyzer like LipiView, it can provide an alternative approach to quantify LLT and batch processing may be executed at the final step with the application software, which is especially beneficial for a clinician or researcher without sufficient budget. We have mentioned the above limitations in the revised manuscript (Lines 366-374, page 12).

Q6 Please specify the ‘macro’ formula in Excel (in another supplementary file) and how you got there.

A6

According to Hwang et al., [11] LLT can be estimated by referring each pixel’s color (Red, green, and blue scales; RGB) to a look-up table (Supplementary file 4) and approximated by the principle of the nearest Euclidean distance. The look-up table of LLT based on Hwang et al. is a 3-dimensional curve in a 3-D Cartesian coordinate system (R axis, G axis, and B axis), and LLT is a determined with 3 variables, R scale, G scale, and B scale. Therefore, after we obtain the 3 scales (r1, g1, b1) of a point (pixel) in an ROI image, we calculated the Euclidean distances between this point and the 408 reference points [(R1, G1, B1), (R2, G2, B2), (R3, G3, B3), …., (R408, G408, B408)] in the look-up table (known RGB scales corresponding to LLTs) according to the distance formula between 2 points in space, distj = [(ri-Rj)2 + (gi-Gj)2 + (bi-Bj)2]1/2. We would then have gotten the 408 distances between this point in the ROI image and the 408 reference points {dist1 = [(r1-R1)2 + (g1-G1)2 + (b1-B1)2]1/2, dist2 = [(r1-R2)2 + (g1-G2)2 + (b1-B2)2]1/2, dist3 = [(r1-R3)2 + (g1-G3)2 + (b1-B3)2]1/2,…., dist408 = [(r1-R408)2 + (g1-G408)2 + (b1-B408)2]1/2}. Finally, we sought out the reference point in the look-up table with the minimal distance and assigned its corresponding LLT as the LLT of this point in the ROI image. The above calculation was repeated in each point of an ROI image. The estimated LLT was determined by averaging LLTs of all points in the ROI image. We have added the formula of Euclidian distance and amended the statement in the paragraph of “Estimation of lipid layer thickness” in the Materials and Methods section (Lines 141-147, page 5).

Round 3

Reviewer 1 Report

In my opinion, the paper is almost ready for publication. Please remember that you don't know the resolution of the method!

Remember (for future publications or presentations of this approach) that you do not know the resolution of this method but this limit should be reported every time!

Author Response

Q1 In my opinion, the paper is almost ready for publication. Please remember that you don't know the resolution of the method! Remember (for future publications or presentations of this approach) that you do not know the resolution of this method but this limit should be reported every time!

A1

We thank the reviewer for this reminder. We have mentioned the limitation in the revised manuscript (Line 360, page 12).